# Energy Harvesting of Deionized Water Droplet Flow over an Epitaxial Graphene Film on a SiC Substrate

**DOI:** 10.3390/ma16124336

**Published:** 2023-06-12

**Authors:** Yasuhide Ohno, Ayumi Shimmen, Tomohiro Kinoshita, Masao Nagase

**Affiliations:** Graduate School of Science and Technology for Innovation, Tokushima University, 2-1 Minamijyousanjima, Tokushima 770-8506, Japan

**Keywords:** graphene, epitaxial, energy harvesting, pure water

## Abstract

This study investigates energy harvesting by a deionized (DI) water droplet flow on an epitaxial graphene film on a SiC substrate. We obtain an epitaxial single-crystal graphene film by annealing a 4H-SiC substrate. Energy harvesting of the solution droplet flow on the graphene surface has been investigated by using NaCl or HCl solutions. This study validates the voltage generated from the DI water flow on the epitaxial graphene film. The maximum generated voltage was as high as 100 mV, which was a quite large value compared with the previous reports. Furthermore, we measure the dependence of flow direction on electrode configuration. The generated voltages are independent of the electrode configuration, indicating that the DI water flow direction is not influenced by the voltage generation for the single-crystal epitaxial graphene film. Based on these results, the origin of the voltage generation on the epitaxial graphene film is not only an outcome of the fluctuation of the electrical-double layer, resulting in the breaking of the uniform balance of the surface charges, but also other factors such as the charges in the DI water or frictional electrification. In addition, the buffer layer has no effect on the epitaxial graphene film on the SiC substrate.

## 1. Introduction

Graphene is a monolayer crystal with carbon atoms in a honeycomb structure that has been intensively investigated for various applications because of its extraordinary carrier mobility and chemical stability [1]. Graphene-based applications, such as electrodes of layered materials [2,3,4], gas [5,6,7,8,9], biomolecule [10,11,12,13,14,15,16] sensors, and optical sources [17,18,19,20], have been intensively investigated.

Over the past decade, the number of sensor devices connected to the Internet has increased. These “Internet of things (IoT)” devices can be used anywhere, such as monitoring a person with a pacemaker, pastured livestock, and traffic control [21]. As the number of IoT devices increases [22], electric power supply becomes difficult, particularly in deserted areas, such as mountains or seaborne areas. However, owing to the adverse effects of carbon dioxide emission on global warming, the importance of various energy-harvesting methods using green, sustainable, and renewable sources, such as sunlight, wind, heat, and vibration, continues to increase. Thus, fossil fuels should not be used as electrical power supplies for these IoT devices in deserted areas. Energy-harvesting devices are required for battery charging. Recently, carbon nanotube networks [23,24,25] and graphene films [26,27,28,29,30,31,32,33,34,35,36,37,38,39,40,41,42] as templates for energy-harvesting devices have been investigated. An electrical potential is generated when a solution droplet flows on the graphene surface. Such energy-harvesting devices can be used in areas with flowing water, such as seashores, rivers, and waterfalls.

However, the origin of such energy-harvesting processes remains unclear. Various mechanisms for nanocarbon-based energy-harvesting devices based on solution flow have been proposed, such as the momentum transfer by phonon [43], breaking of the uniform balance of the surface charges [27], and fluctuation of asymmetric potential [44]. However, the mechanism of energy harvesting from solution flow has not yet been determined. Recently, the relationship between the electrode pattern and solution flow direction as well as the importance of the existence of ions have been reported [37,45]. Although the crystal quality, surface, and substrate conditions of graphene are considered crucial, most studies used a graphene film synthesized by chemical vapor deposition (CVD), which has many defects and dislocations owing to the polycrystalline metal foil catalyst. Thus, defect-free, large, single-crystal graphene films with no polymer residues are required to understand these physical mechanisms.

We have investigated the synthesis and application of a large-area single-crystal epitaxial graphene film on a 4H-SiC(0001) substrate. Epitaxial graphene field-effect transistor (FET)-based sensors have unique characteristics. For example, epitaxial graphene FET cannot detect conventional ions [46], indicating that the surface of epitaxial graphene film is covered by ideal π-orbitals of sp^2^ bonds without defects and dislocations. Furthermore, protein and amino acid adsorption on the epitaxial graphene film generally shows electron doping [47,48]. Therefore, epitaxial graphene on a SiC substrate can be expected to obtain almost ideal graphene-based FET sensor applications because of the defect- and dislocation-free graphene crystals. Therefore, it is very important to show various energy harvesting data using the epitaxial graphene film on the SiC substrate to clarify the mechanism.

This study investigates epitaxial graphene-film-based energy harvesting characteristics using deionized (DI) water droplets. We investigate the dependence of the flow direction on the electrodes to understand the role of the electrical-double layer. Another carbon layer (buffer layer) exists under the graphene layer in the epitaxial graphene film on the Si face of the 4H-SiC(0001) substrate whose structure is the same as that of the graphene film; however, the buffer layer is connected to the SiC surface via covalent bonds [49]. Covalent bonds are one of the origins of the high electron doping (>5 × 10^12^ cm^−2^) to the epitaxial graphene film. The buffer layer is considered an electrically inactive film owing to the lack of sp^2^ bonds and the presence of numerous covalent bonds. The influence of the buffer layer should be clarified for epitaxial graphene film-based devices. Hence, we investigated the buffer-layer dependence.

## 2. Materials and Methods

A semi-insulating 4-inch 4H-SiC(0001) wafer was purchased from Cree, Inc. The wafer was diced into 10 × 10 mm^2^ chips using the stealth dicing method and used as the substrate. After the SiC substrates were cleaned using hydrofluoric acid, piranha solution (H_2_SO_4_ + H_2_O_2_), and DI water, they were placed in a rapid thermal annealer (SR1800, Thermo Riko Co. Tokyo, Japan) and annealed at 1620 °C for 5 min in an Ar atmosphere at 100 Torr. The epitaxial graphene films used in this work were almost covered by the monolayer graphene. The average sheet resistance, electron mobility, and electron concentration of the epitaxial graphene films were 900 Ω/sq., 850 cm^2^/(Vs), and 7 × 10^12^ cm^−2^, respectively, as measured using the van der Pauw method, indicating that the crystal quality of the epitaxial graphene film was sufficiently high [50]. The quality of the graphene crystals was evaluated by Raman spectroscopy (uRaman, Technospex Co. Singapore).

An epitaxial graphene film on a SiC substrate was placed on a tilted Macor^®^ stage with an angle of 45°. Cu probes were used, as shown in Figure 1a,b. DI water was dropped on the epitaxial graphene surface using a micropipette. In this study, we measured the voltage generated using two electrode configurations—that is, parallel and perpendicular patterns to the DI water flow direction, as shown in Figure 1c,d. The signal was amplified using a differential amplifier 5307 (NF Corp. Yokohama, Japan) and collected using a digital oscilloscope DSO6014A (Keysight Technologies Inc. Santa Rosa, CA, USA). Voltage generation measurements using DI water flow were performed 50 times for both parallel and perpendicular electrode configurations.

The buffer layer under the epitaxial graphene film was electrically activated via hydrogen intercalation. The Si surface of the SiC substrate is terminated by the hydrogen atoms. Consequently, the buffer layer became electrically active—that is, a graphene layer called “quasi-free-standing graphene” [49]. Therefore, the monolayer epitaxial graphene film became a bilayer epitaxial graphene film without a buffer layer after hydrogen intercalation. In this study, the epitaxial graphene film was annealed in an electrical furnace at 1000 °C for 60 min with Ar/H_2_(4%) in atmospheric pressure. The characteristics of the epitaxial graphene film were investigated using van der Pauw–Hall measurements and Raman spectroscopy after H_2_ intercalation.

## 3. Results

The highest average voltage responses of the parallel and perpendicular electrode configurations for over 50 measurements are shown in Figure 2a,b. The voltage precipitously increased within approximately 70 µs and then decreased within approximately 350 µs. Every generation event terminated within 500 µs, which was much shorter than those in other studies. Because the water droplet fell from the substrate within approximately 100 ms in this measurement, which was validated by high-speed camera images, the event time of approximately 500 µs was quite short. Although the short event time was considered to be derived from the size of the substrate of 100 mm^2^, which was smaller than those of other studies [27,30,35,37,41], further studies on the tilt angle or fluid velocity dependences to clarify the short event time are required.

Remarkably, the voltage values generated by the DI water flow were as high as over 50 mV. The maximum voltage generation was greater than 100 mV for the parallel and perpendicular electrode configurations. In previous studies, using CVD-grown graphene films, the voltage generated by the DI water flow was quite small [28,45], and transition metal dichalcogenide-based energy harvesting devices showed no voltage generation [51]. Therefore, energy harvesting by solution flow requires ions, such as Na^+^ and Cl^−^. However, energy can be harvested from DI water flow in epitaxial graphene films on SiC substrates.

The results of 50 measurements of voltage generation by the DI water flow on the epitaxial graphene film for the parallel and perpendicular electrode configurations are shown in Figure 3a. The solid circles represent the average values, and the error bars represent the standard deviations. Although the voltage generated by the DI water flow for the perpendicular electrode configuration was slightly larger than that of the parallel electrode configuration, the values of both voltages generated were similar, which differed from those of previous studies [37,45].

The resistance of the epitaxial graphene film was measured using the conventional two-terminal method to evaluate the electrical power of the energy-harvesting device. The current (*I*)–voltage (*V*) characteristics of the parallel and perpendicular electrode configurations are shown in Figure 3b. The red [blue] solid lines represent the parallel [perpendicular] electrode configuration. The *I*–*V* characteristics were linear, indicating that the contact between the Cu probes and epitaxial graphene film was Ohmic. The inset shows the differential resistance of the *I*–*V* characteristics. The average resistance was 2.7 and 3.0 kΩ for the parallel and perpendicular electrode configurations, respectively. The average sheet resistance of the epitaxial graphene film was 0.9 kΩ/sq., and the contact resistance was approximately 1.2–1.4 kΩ, which was quite lower than those of other graphene contact resistances [52]. The difference between the parallel and perpendicular electrode configurations was derived from the terrace and step structures [53] of the 4H-SiC(0001) surface used in this study.

The electrical power was obtained as the average differential resistance and voltage (*V*^2^/*R*, where *R* and *V* are the average differential resistance and generated voltage, respectively). The average electrical power of the parallel and perpendicular electrode configuration was 1.2 and 1.6 µW, respectively, which was quite large for other graphene-based energy harvesting devices using DI water or low ion concentration solution flow [35,37]. Suppose the electric power is generated in 500 µs. In this case, the electrical energy corresponded to approximately 10 Wh.

Typical Raman spectra of the epitaxial graphene film before and after H_2_ intercalation are shown in Figure 4a. Several peaks were observed in the Raman spectrum of the as-grown epitaxial graphene film ranging from 1200 to 1500 cm^−2^. These peaks were derived from the buffer layer [54] and not from the D peak, which was because of defects and dislocations. After H_2_ intercalation, the peaks from the buffer layer disappeared, and a D peak appeared. The origin of the appearance of the D peak can be considered as the transformation from the buffer to the graphene layer. In general, several buffer layer areas of approximately 10^−2^ µm^2^ sometimes appeared in the epitaxial graphene film on the SiC substrate [50]. The formation of a boundary between the mono- and bilayer graphene resulted in the D peak. In addition, the full width at half maximum (FWHM) of the 2D peak increased after H_2_ intercalation. The FWHM of the 2D peak for the epitaxial graphene film before H_2_ intercalation process was 35.6 cm^−2^, which was the conventional value for the epitaxial monolayer graphene film. It increased to 55.8 cm^−2^ after H_2_ intercalation. This increased FWHM value indicates the transformation from a monolayer to a bilayer graphene film by hydrogen termination under the buffer layer. Moreover, Hall-effect measurements showed a clear change in carrier type. Before [after] H_2_ intercalation, the sheet resistance, carrier concentration, mobility, and carrier type were 710 [2000] Ω/sq., 8.9 × 10^12^ [1.2 × 10^13^] cm^−2^, 980 [250] cm^2^/(Vs), and n-type [p-type], respectively. The carrier-type change is typical of H_2_ intercalated epitaxial graphene films [49]. These results demonstrate the fabrication of a quasi-free-standing epitaxial graphene film.

A summary of the voltage generated by 50 measurements using DI water flow is shown in Figure 4b. The average generated voltages for the epitaxial graphene film after H_2_ intercalation for the parallel and perpendicular electrode configurations were 63 and 46 mV, respectively, which were similar to those before H_2_ intercalation, indicating that the buffer layer under the epitaxial graphene film was not influenced by energy harvesting from the solution flow.

The dependence of the generated voltage on the droplet amount in the epitaxial graphene film with the buffer layer is shown in Figure 5. Initially, the generated voltage increased with increasing the droplet amount and later saturated. The generated voltage linearly increased with an increase in the volume of droplet, indicating that the charges in the DI water droplet or friction between graphene and DI water droplets might be influenced.

## 4. Discussion

As shown in Figure 2 and Figure 3, the epitaxial graphene-based energy-harvesting device showed large voltage (electrical power) generation using DI water flow; this energy-harvesting effect showed almost no dependence in the flow direction. In CVD-grown graphene film-based energy-harvesting devices, the voltages generated by DI water flow were quite small (several millivolts) [35,37], and a clear flow direction dependence could be observed. Therefore, the mechanisms of energy harvesting by DI water flow differ between the epitaxial and CVD-grown graphene films.

The origins of energy harvesting by solution droplets flowing on the graphene film differ as mentioned in the introduction. A key interpretation is the significance of the existence of anions and cations as well as the broken uniform balance of surface charges in graphene by the electrical double layer in the moving solution droplet [27]. Therefore, the voltage generated by the solution flow showed the ion concentration and electrode configuration dependence on the generated voltage in CVD-grown graphene films. However, the generated voltages were observed under DI water flow, and electrode configuration dependence was not observed in the epitaxial graphene film.

Because epitaxial graphene films on the SiC substrate used in this study were single crystals with large areas with no polymer residue, ideal results were expected for graphene-based devices. Therefore, the energy harvested by the solution flow on the graphene surface must not be anions or cations in the solution. Hence, the electrical double layer is inessential in this case.

The generated voltages depended on the number of DI water droplets as shown in Figure 5. Because DI water (ultrapure water) is charged [55,56], the charges in DI water might be influenced by the generated voltages. Furthermore, friction between the graphene surface and the DI water droplet may cause frictional electrification [57,58]. However, the generated voltages were almost saturated at 20 µL; hence, further studies on the ion concentration and droplet flow velocity dependence are required. In addition, the dynamics from the impact to the dropping down of the droplet are crucial; hence, time-course measurements using a high-speed camera are also required to clarify the mechanism of energy harvesting with DI water flow on the epitaxial graphene film.

## 5. Conclusions

This study investigated energy-harvesting characteristics of single-crystal epitaxial graphene films synthesized on SiC substrates using DI water flow. A voltage as high as 100 mV was generated by the DI water flow, and the generated voltage was independent of the electrode configurations parallel and perpendicular to the DI water flow as well as the existence of the buffer layer under the epitaxial graphene film. The average electrical power of energy harvesting by the DI water flow exceeded 1 µW. The epitaxial graphene film-based energy-harvesting devices using solution flow can be used for electrical power supply to IoT devices in the presence of flowing water, such as river or seashore, with no electrode configuration in the direction of water flow.

## Figures and Tables

**Figure 1 materials-16-04336-f001:**
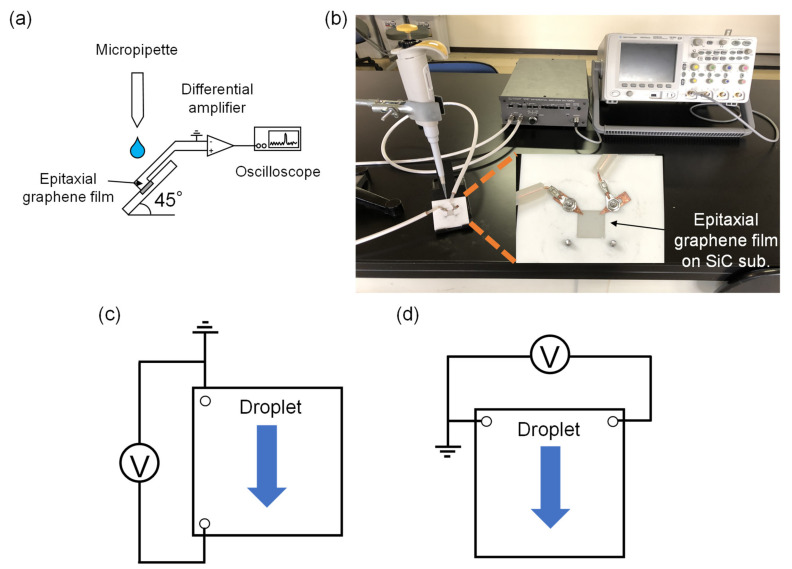
(**a**) Schematic of the measurement setup. (**b**) Photograph of the measurement, which was the perpendicular electrode configuration to the direction of DI water flow. (**c**,**d**) Parallel and perpendicular electrode configurations to the direction of DI water flow, respectively.

**Figure 2 materials-16-04336-f002:**
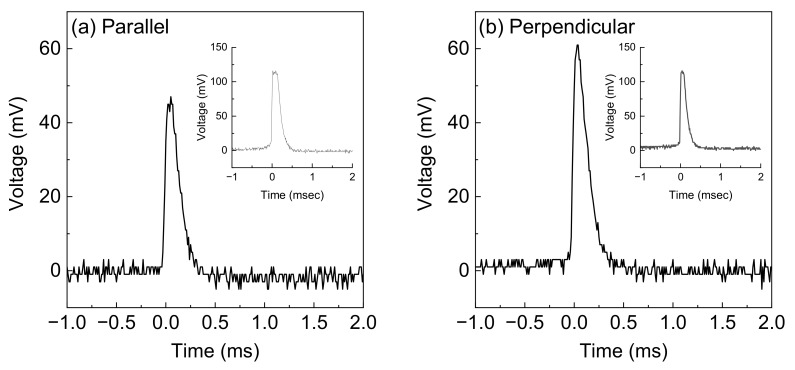
Average voltage response of (**a**) parallel and (**b**) perpendicular electrode configurations. The insets show the voltage response at the maximum case.

**Figure 3 materials-16-04336-f003:**
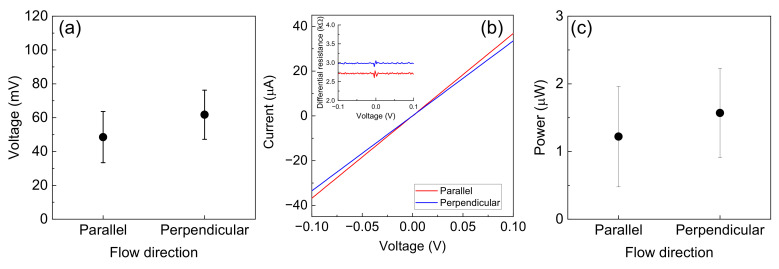
(**a**) Summary of the voltage generated by 50 measurements using DI water flow. The solid circles and error bars represent the average values and standard deviation, respectively. (**b**) *I*–*V* characteristics of the two-terminal measurement for the parallel and perpendicular electrode configuration. The inset shows the differential resistance (*dV*/*dI*). (**c**) Generated electrical power (*V*^2^/*R*) for the parallel and perpendicular electrode configuration. The solid circles and error bars represent the average values and standard deviation, respectively.

**Figure 4 materials-16-04336-f004:**
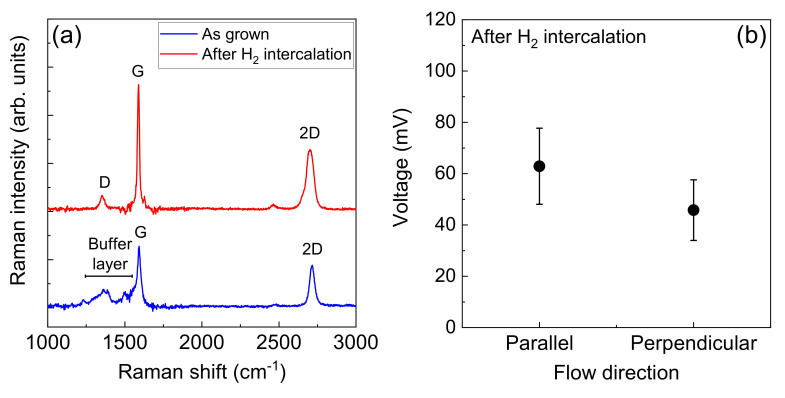
(**a**) Typical Raman spectra of the epitaxial graphene film before and after H_2_ intercalation. The D, G, and 2D peaks, which are the distinctive peaks from a graphene film, can be observed at around 1350, 1590, and 2680 cm^−1^. (**b**) Summary of the voltage generated by 50 measurements using DI water flow for the epitaxial graphene film after H_2_ intercalation.

**Figure 5 materials-16-04336-f005:**
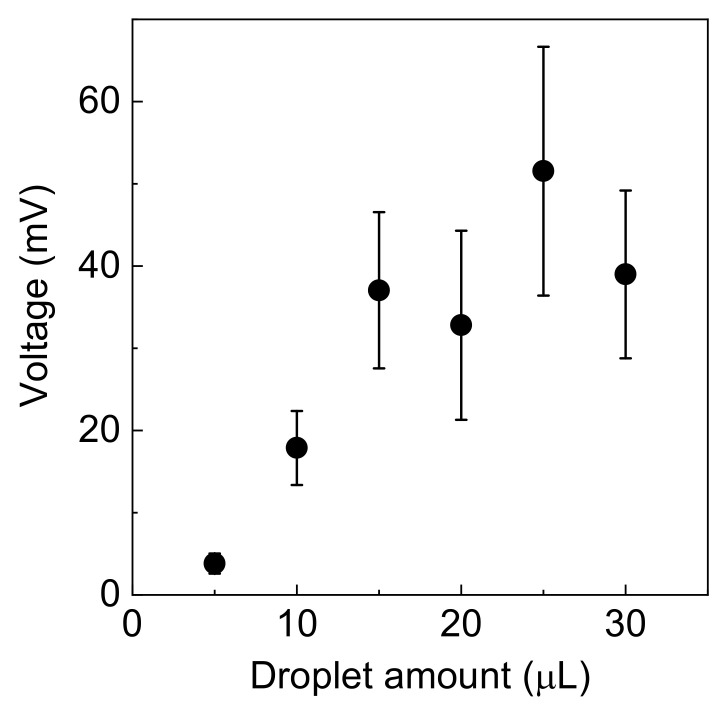
The dependence of the generated voltage on droplet amount. The solid circles and error bars represent the average values and standard deviation, respectively.

## Data Availability

The data presented in this study are available on request from the corresponding author.

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
