# Peer review of "Energy Harvesting of Deionized Water Droplet Flow over an Epitaxial Graphene Film on a SiC Substrate"

_materials, 2023, doi:10.3390/ma16124336_

Round 1
Reviewer 1 Report
The current study looks interesting for publishing after considering some points below:
- The abstract should be enhanced by introducing some of your results and findings.
- Can you think about Si or cheaper substrates instead of SiC for more commercialization purposes? Is it possible to use another substrate than SiC? What do you expect
- May you add some information about the preparation Graphene, how many layers?
- Fig 3(b), can you use an mV unit
- Fig 4 (b), is the unit in mV or V ? Also for Fig 5, keep all in mV better
- What do you think about using other 2D materials such as MoS2, Or WS2 instead of using Graphne? can you add a small paragraph about it by the end of the manuscript explaining the potential of using it? This ref may help you
https://www.researchgate.net/profile/Mohammed-Tihtih-2/publication/364322534_Nanostructured_MoS2_and_WS2_Photoresponses_under_Gas_Stimuli/links/634801f0ff870c55ce215d2a/Nanostructured-MoS2-and-WS2-Photoresponses-under-Gas-Stimuli.pdf
some minor corrections needed
Author Response
Please check the pdf fils.

Reviewer 2 Report
The authors report on energy generation by droping water droplets onto epitaxial graphene formed on SiC substrates. The authors conduct two configurations in which the direction of droplet flow is either parallel or perpendicular to the electrode direction. The generation of electricity on graphene has been already published in 2014 (Nature Nanotechnology 9, 378–383 (2014)) so the concept is not novel. However, the use of DI water droplet along with epitaxial graphene formed on SiC is rather interesting. However, the presentation and discussion needs to be improved significantly for publication. Here are some major points to improve:
1. What's the purpose of trying different electrode configuration? Although the energy generated is slightly different (almost no difference considering the spread of the data), there are no real physical explanation or reason for this experiment. Also, there could be many other factors other than the direction of the electrode for the different power generation, which are not discussed.
2. It is difficult to compare and understand the difference between H2 intercalated results and not intercalated results. Figure 4 and 5 show voltage generation in the V scale (reviewer assume it is a typo). If there are difference between intercalated and non-intercalated results, what is the physical reason? Are the difference significant enough to warrant investigation?
3. The authors only carry out 50 trials for each electrode configuration, which seems like a very limited number of trials considering the ease of experiment. The spread of the data is also very large (considering figure 5), which seems like any conclusion derived from these data could be just due to the variation in measurement. The authors should conduct more iterations (maybe 200+) to get a more accurate idea of the data.
4. There is a lack of any physical understanding or explanation of the results and only speculation.
Some minor points:
1. The figures are not very professional and the scale seems to have typos on some of them (Figure 4, 5 y-axis).
2. There are grammatical errors as well as wrong sentences (check last sentence of conclusion section).
Few grammar mistakes, and at least one wrong sentence.
Author Response
Please check the pdf file.

Round 2
Reviewer 1 Report
Thank you for your report.
some more revisions are needed
Reviewer 2 Report
The authors have made sufficient edits to the manuscript. I guess the manuscript can be published as is.